# Validation and Analysis of Recreational Runners’ Kinematics Obtained from a Sacral IMU

**DOI:** 10.3390/s25020315

**Published:** 2025-01-07

**Authors:** Hossein Asgari, Ben Heller

**Affiliations:** Sport and Physical Activity Research Centre, Sheffield Hallam University, Olympic Legacy Park, 2 Old Hall Rd, Sheffield S9 3TY, UK; hasgari2@uwo.ca

**Keywords:** running biomechanics, running kinematics, running movement patterns, inertial measurement unit, wearable sensor, clustering runners, running performance, running injury

## Abstract

Our aim was to validate a sacral-mounted inertial measurement unit (IMU) for reconstructing running kinematics and comparing movement patterns within and between runners. IMU data were processed using Kalman and complementary filters separately. RMSE and Bland–Altman analysis assessed the validity of each filtering method against a motion capture system. Running data from 24 recreational runners were analyzed using Fourier transform coefficients, PCA, and k-means clustering. High agreement was found for Kalman-filtered data in the frontal, sagittal, and transverse planes, with a Bland–Altman bias of ~2 mm on average, compared to a bias of ~10.5 mm for complementary-filtered data. Pelvic angles calculated from Kalman-filtered data had superior agreement, with systematic biases of ~0.3 versus 3.4 degrees for complementary-filtered data. Our findings suggest that inertial sensors are viable alternatives to motion capture for reconstructing pelvic running kinematics and movement patterns. In the second part of our study, negligible intra-individual differences were observed with changes in speed, while inter-individual differences were large. Two clusters of runners were identified, each showing distinct movement patterns and ranges of motion. These observations highlight the potential usefulness of inertial sensors for performance analysis and rehabilitation as they may permit the use of individual-specific and cluster-specific practice programs.

## 1. Introduction

Running technique is one of the most important components in studying running efficiency and performance [1]; additionally, improper technique increases the risk of injury [2,3,4].

Since individual runners are unique, general recommendations regarding running biomechanics do not necessarily apply to all [5]. Therefore, individual- and/or cluster-specific analyses may provide more accurate and beneficial feedback. Researchers usually classify runners into different groups according to their sex, age, experience/skill level and health to compare biomechanics [6]. However, even amongst skilled performers, running patterns differ [7,8]. These are important factors to avoid incorrect interpretation when studying running technique in different people. Clustering different running patterns could identify movement patterns associated with better performance or higher risk of injury, while specific practice programs could be developed for each cluster.

Studying running kinematics is necessary to identify an individual runner’s technique and subsequently identify clusters. Phinyomark et al. [8] used an optical motion capture system and focused on lower limb joint kinematics to cluster runners. However, they revealed no information about running symmetry, which is an important factor in running since it could be an indicator of injuries or differences in running techniques [9]. Mezghani et al. [10] classified runners according to their knee joint kinematics, only considering the frontal plane in their study; therefore, their classifications could not fully describe running movements. These examples highlight the need for studying running movement patterns and performing cluster analysis, while considering the motion in all planes.

Running kinematics are typically captured using optical motion capture systems [11]. However, these systems are not only expensive and limited in availability, but they are typically used indoors whilst running on treadmills. Therefore, they provide data which may not represent true practice. As a consequence, the relatively cheap, lightweight and fully portable devices known as inertial measurement units (IMUs) have become popular alternatives to optical systems [12,13,14].

The IMU’s accelerometer data are noisy and influenced by both orientation and linear acceleration. Their rate-gyroscopes measure angular velocities around each axis, but the integrated orientation angles are subject to drift over time. Therefore, sensor fusion, which combines both types of sensor readings to compensate for their individual disadvantages, is necessary for accurate orientation sensing. The most popular sensor fusion methods are the complementary filter and the Kalman filter [15].

Since the sacrum’s location is close to that of the COM [16,17], its movement is often used as a proxy for COM movement. Therefore, taking measurements at the body’s center of mass (COM) is beneficial as it gives insight into the overall biomechanics of the body.

The primary purpose of this study was to validate the use of a sacral-mounted IMU for capturing running kinematics, comparing Kalman and complementary filtering methods. The secondary purpose was to capture sacral kinematics for a wide range of participants, and to investigate if any common patterns (clusters) could be defined.

## 2. Materials and Methods

### 2.1. Validation Study

#### 2.1.1. Data Collection

Raw data were collected originally by Jasmin Willer at Sheffield Hallam University for her master’s report “Characterization of running movement patterns using a single inertial sensor at the sacrum”. In the current study, a secondary data analysis was performed on her data.

Six participants were recruited for the validation stage of the study (Table 1). A triangular marker-cluster was attached to each participant’s sacrum using double-sided tape, with an IMU mounted in the middle of the triangle using a belt and flexible bandage (Figure 1). The 25 g IMU (Opal, APDM, Portland, OR, USA) consisted of an accelerometer (±6 g), a gyroscope (±2000 g) and a magnetometer (±6 Gauss). Data were captured at 128 samples/s; however, the magnetometer was disabled as it would be affected by the metal in the treadmill and environment. A Motion Analysis Corporation eight-camera optical motion capture (MOCAP) system (Santa Rosa, CA, USA) was used to capture the 3D marker positions at 200 Hz.

All participants completed four 30 s trials at their estimated 1 km, 5 km, half-marathon and marathon speeds to cover various running speeds for validation. Their individualized speeds were calculated based on their age-graded running score and personal 5 km times provided by the participants.

#### 2.1.2. Data Analysis

All marker data were filtered using a second-order band-pass bidirectional Butterworth filter with a low cut-off frequency of 0.5 Hz to remove drift in the participant’s position and a high cut-off of 10 Hz [16]. The motion capture data were resampled to 128 Hz to match the sampling rate of the IMU. The angular displacements of the pelvis were calculated with Visual 3D (C-motion Inc., Germantown, MD, USA) and presented as Euler angles with a rotation sequence of rotation, obliquity, tilt (ROT) following Baker (2001) [18]. All other data processing was conducted using MATLAB (MathWorks Inc., Natick, MA, USA). For the motion capture system, a right-handed coordinate system was used with the location of the IMU calculated as the midpoint of the triangle. Finally, the displacement data were differentiated twice using a five-point central finite difference method to obtain velocity and acceleration.

As the IMU’s raw acceleration data contain the apparent acceleration due to gravity as well as true coordinate acceleration, they need to be processed before further analysis. By using sensor fusion approaches, the sensor orientation is calculated and used to remove gravity for each trial [19].

The complementary filter combined the accelerometer and gyroscope readings using the equation below [20].
(1)$$Angle\left(t\right)=0.985*\left(Angle\left(t-1\right)+gyro\left(t\right)*\frac{1}{128}\right)+{Acc}_{Angle\left(t\right)}*0.015$$

“*Angle*” is the obtained angle from combining the accelerometer and gyroscope data at every time-step “*t*”.

“*Acc_Angle(t)_*” is the angle calculated using accelerometer data, “*gyro(t)*” is the angular velocity obtained from the gyroscope, and the numbers 0.985 and 0.015 are the filter values for the high-pass and low-pass filters, respectively, chosen by visual examination. Equation (1) was used to obtain obliquity and tilt angles. To calculate obliquity and tilt using accelerometer data “*Acc_Angle(t)_*”, the signals were first low-pass filtered using a second-order Butterworth filter with a cut-off frequency of 1.5 Hz to reduce noise. Tilt (theta) and obliquity (psi) were then calculated using the following equations [19].
(2)$$\theta ={tan}^{-1}(-\frac{{Acc}_{x}}{\sqrt{{{Acc}_{y}}^{2}+{{Acc}_{z}}^{2}}})$$


(3)
$$\psi ={tan}^{-1}(-\frac{{Acc}_{z}}{\sqrt{{{Acc}_{y}}^{2}+{{Acc}_{x}}^{2}}})$$


Rotation of the pelvis was calculated by integrating the gyroscope signal, and to correct for integration drift, the data were subsequently filtered using a second-order high-pass Butterworth filter with a cut-off frequency of 0.2 Hz, chosen by observation. Using the obtained rotation matrix, gravity was subtracted from the raw acceleration data at each time-step. The processed vertical and medio-lateral acceleration data were low-pass filtered with a cut-off frequency of 10 Hz, which was found to be the best cut-off frequency to obtain accurate peak accelerations [21]. For anterior–posterior acceleration a cut-off frequency of 15 Hz was found to be more accurate through observation. The acceleration data were integrated and doubly integrated to calculate the velocity and displacement, respectively. To correct for integration drift, a second-order high-pass Butterworth filter was then used with a cut-off of 1 Hz for all directions found by observation and comparison with the MOCAP system results.

For the Kalman filter, the accelerometer and gyroscope readings were first aligned with the north-east-down coordinate system (NED). Simulated north-pointing magnetometer data were created to remove orientation drift. Afterwards, the “ahrsfilter” MATLAB function was used to fuse the data and output the sacrum’s rotation matrix. The rotation matrix was used to subtract gravity from the acceleration data. To obtain the sacrum’s angles, the function’s output was converted from quaternion to Euler angles using the rotation sequence ROT. The acceleration data were integrated and double integrated to calculate the velocity and displacement, respectively. To correct for integration drift, a second-order high-pass Butterworth filter was subsequently used with filter values being found by observation versus the MOCAP system. The best filter values were 0.5 Hz for the medio-lateral axis and 0.8 Hz for the other two axes.

The following process was performed for the filtered data obtained by each filtering method. First, the data for the first and last five seconds of the trials were removed to ensure consistent steps. To synchronize the IMU and MOCAP system data, cross-correlation and relative lag between the two measurement systems were calculated and used to align and crop each trial accordingly. Instantaneous RMSEs of accelerations, velocities, displacements, and angles were calculated for each trial and each axis, comparing every single data point obtained from the IMU to the MOCAP system. Then, the means of the obtained RMSEs were calculated across trials for each individual, and subsequently across all the participants, as a measure of instantaneous accuracy. To evaluate the validity and reliability of the IMU-derived data for capturing mean kinematics of runners across their strides (e.g., overall movement patterns), we first defined step and stride cycles. Peak accelerations in the vertical direction were used to define step cycles in each trial [22]. Subsequently, every step and stride were interpolated to 100 and 200 data points, respectively. Afterwards, the mean accelerations, velocities, displacements, and angles were calculated over each step and stride. The mean trajectories were plotted, while their ranges were calculated and stored for further analysis. To validate the IMU against the MOCAP system, Bland–Altman plots were used to compare the obtained ranges [23]. Additionally, RMSEs of accelerations, velocities, displacements, and angles were calculated for the mean kinematics of each trial. Then, the means of the obtained RMSEs were calculated across trials for each individual, and subsequently across all the participants.

### 2.2. Movement Pattern Study

#### 2.2.1. Data Collection

For the second stage of the study, only the IMU was attached on the participants’ sacrum. Twenty-four participants of different running abilities were recruited, with each completing three 5 min trials at their individualized 5 km, 15 km, and marathon speeds, calculated as explained before. A 1% inclination was applied to the treadmill for all tests to compensate for the lack of air resistance occurring in real over-ground running. Rest time was provided between trials. Both data collections obtained ethical approval from Sheffield Hallam University.

#### 2.2.2. Data Analysis

The more accurate filtering method (Kalman filter) was used to process the running data of the 24 participants, while the first and last 10 steps were removed to ensure a consistent gait. The steps and strides were identified and interpolated to 100 and 200 data points, respectively. Finally, the mean acceleration, velocity, displacement, and angle plots were produced for each trial. MATLAB’s curve fitting tool was used to model each axis as a Fourier series. The curves were reconstructed with good accuracy using 17 terms in the series (a constant, 8 sine and 8 cosine terms). The Fourier coefficients for all axes and for all the selected trials were calculated and stored.

Principal component analysis (PCA) was used to reduce the dimensionality of the data. Using the principal components (PCs) and a k-means clustering method, the major clusters and their corresponding participants were identified. A two-sample *t*-test was performed to determine whether the differences between the groups were statistically significant. The effect sizes of these variables were calculated using Cohen’s d. Finally, to compare the male and female proportions between the two groups and to investigate if the clusters were different in gender, a Chi-squared test was used.

## 3. Results

### 3.1. Validation Study

To compare IMU-measurements against the gold standard measurements for each filtering method, instantaneous and mean stride RMSEs as well as Bland–Altman plots with 95% limits of agreement for accelerations, velocities, displacements, and angles were produced. The summary of these analyses is shown in Table 2, and the displacement Bland–Altman plots are shown in Figure 2. For visual inspection of the accuracy of the Kalman filtering, the movement patterns of one participant produced using each of the measurement systems are illustrated in Figure 3.

### 3.2. Movement Pattern Study

In the second stage of the study, the movement patterns of every participant in the frontal, sagittal and transverse planes were reconstructed for each speed. Movement patterns for two participants are shown in Figure 4 as examples. Subsequently, all the movement patterns were quantified as described above, and their Fourier coefficients were stored in a 24 × 51 matrix. Performing the principal component analysis (PCA) revealed that by using 3, 5, 10 and 23 principal components, 76%, 90%, 98% and approximately 100% of the variance of the data were explained, respectively. To ensure the highest reconstruction quality and optimal cluster separability in our dataset, all PCs were retained for subsequent analysis [24]. Using all 23 PCs and the k-means clustering method, two clusters and eight individuals were identified (Figure 5). The first and second clusters comprised twelve and four participants, while the other eight individuals were not close enough to create more clusters. The mean coefficients for each cluster were used to reconstruct the Fourier coefficients, and hence, the mean displacements in frontal, sagittal and transverse planes were plotted for each cluster in Figure 6. To compare ages, heights, weights and speeds between the two clusters, the *t*-test results show that the *p*-values were all above 0.05 except for weight, which had a *p*-value of 0.04 (Figure 7). The second lowest *p*-value was for height (*p* ≈ 0.12). The effect sizes for age, height, weight and speed were 0.09, 1.1, 1.3 and 0.4, respectively. Finally, Chi-square test results for gender indicate a Chi-square statistic of 0.35, with *p*-value of 0.55.

## 4. Discussion

### 4.1. Validation Study

Our first aim was to validate the sacral-mounted IMU against the motion capture system and to compare the complementary and Kalman filtering methods for estimating instantaneous and overall kinematics from the IMU outputs. For calculated accelerations, results of the Bland–Altman analysis showed higher agreement for the Kalman filter with the gold standard method than for the complementary filter (Table 2). The ranges between the upper and lower limits of agreement of acceleration in the anterior–posterior (AP) and medio-lateral (ML) directions were significantly smaller for the Kalman filter. For acceleration in the vertical direction, there was a similar agreement for both filtering methods against the MOCAP system results. The mean difference (systematic bias) of the Kalman filter was smaller for ML and vertical directions. Moreover, the RMSEs of accelerations in all three directions were lower for the Kalman filter, for instantaneous, as well as mean stride accelerations. For velocities, the systematic biases as well as the RMSEs of instantaneous and mean stride velocities were lower for the Kalman filter in all directions, while the bias of the complementary filter was relatively high (Table 2). Similarly, the systematic biases, as well as the instantaneous and mean stride RMSEs of the Kalman filter were lower for displacements on all axes (Figure 2 & Table 2). The overall movement patterns of individuals were captured with high accuracy by the Kalman-filtered IMU data, with the average bias of ~1.9 (mm) across all axes, compared to a bias of ~10.5 (mm) for the complementary-filtered data. The mean stride displacement RMSEs were only 3.7, 0.8, and 1.8 (mm) on AP, ML, and vertical axes, while the respective instantaneous RMSEs were 5.2, 3.1, and 5.8 (mm). This highlights the even higher reliability of Kalman-filtered IMU data when capturing overall movement patterns of runners, compared to instantaneous stride-to-stride measurements. Given the low values of RMSE of the mean gait patterns compared to the ranges of motion in each axis, we can be confident that these mean traces are a valid representation of each individual’s movement patterns.

For the pelvis angles, the systematic biases, as well as the ranges of the limits of agreement were unacceptably high for the complementary filter (Table 2). On the other hand, good agreement was found for the angles obtained using the Kalman filter, with considerably smaller systematic biases and limits of agreement. Furthermore, the instantaneous RMSEs of pelvis angles for obliquity, tilt, and rotation were only 0.85, 1.06, and 1.54 degrees, respectively, indicating the validity of Kalman-filtered IMU data for capturing instantaneous pelvis angles (Table 2). Consistent with our findings for linear movements, the mean stride pelvis angles, compared to instantaneous pelvis angles, indicate an even higher reliability, with obliquity, tilt, and rotation RMSEs of only 0.77, 0.69, and 1.22 degrees. Earlier studies indicate that during gait and running, errors below 5 degrees are considered satisfactory for clinical and performance interpretations [25,26]. In our study, the RMSEs for instantaneous and overall pelvis angles calculated using the Kalman filter were considerably lower than this suggested threshold, indicating the accuracy and reliability of the sacral IMU, and this specific sensor fusion method for capturing pelvis angles.

Based on the analysis above, the Kalman filter is the preferred and more accurate filtering approach to calculate accelerations, velocities, displacements and pelvis angles using a sacral IMU. Additionally, the performance of the Kalman filter is even better when capturing overall kinematics of runners. Visual examinations also confirmed the very good accuracy of the Kalman filtering approach. As shown in Figure 3, the overall shapes of the patterns as well as the ranges of motions (ROMs) are very close between the two systems and in all planes. Although this study demonstrated better accuracy and validity for the Kalman filter compared to the complementary filter, it is possible that alternate filter parameters for the complementary filtering method may have improved its performance. Finally, it can be stated that IMUs are capable of capturing sacral movements and pelvis angles with good accuracy and are therefore valid alternatives to optical motion capture systems, especially for reconstructing overall sacral kinematics. It should be noted that even though performance was good, the IMUs used in the study were relatively heavy at 25 g, and had a limited sampling rate of 128 samples/s. Higher-performance IMUs might improve the results further.

### 4.2. Movement Pattern Study

The second aim of the study was to analyze the movement patterns of the 24 participants. The movement patterns of two participants for different speeds are shown in Figure 4; two important features can be seen immediately. Firstly, although small intrapersonal differences in ranges of motion were associated with changes in running speed, especially in the medio-lateral direction, the overall shape of the movement patterns in all planes remained similar for each participant in all planes. This was apparent for all participants, especially the experienced runners, where it was difficult to visually identify changes in their running patterns with changes in their speed. Therefore, regardless of running speed, each participant had a distinctive pattern in each plane. Secondly, the movement patterns and ranges of motions differed between the two participants in all planes (Figure 4). This uniqueness was apparent for all participants, and is consistent with Phinyomark et al.’s findings of differences within a group of healthy runners [8].

The mean movement patterns of the clusters illustrated in Figure 6 show that the two clusters differed in the shapes of their movement patterns as well as their ROMs in all planes. The second cluster had a larger range of medio-lateral and anterior–posterior displacement. Moreover, considerable differences in anterior–posterior ROM were found between the right and left steps in cluster 2. The biggest asymmetries can be found in cluster two and specifically in sagittal and transverse planes, while the members of cluster 1 produced relatively more symmetric patterns. Looking at the mean 5 km speeds in the two clusters as an indicator of performance, the second cluster was slightly faster with mean 5 km speed of 3.47 m/s (12.13 km/h), versus 3.29 (11.84 km/h) for the second cluster. Comparing the two groups of participants in Figure 7, *t*-test results and effect sizes, the two groups were different in height and weight with large effect sizes of 1.1 and 1.3, respectively. The *t*-test results also suggested that the two clusters were different in weight, with a *p*-value of 0.04 (Figure 7). The *t*-test results, effect sizes, and Chi-square statistics suggested that the two clusters were not significantly different in age, speed, or gender. However, these comparisons and test results may be more reliable if the two clusters were larger, especially for the second cluster, which only had four members. A larger number of participants should be recruited in future research to more reliably compare different variables between clusters.

As the shape of the movement patterns appear distinctive for each person regardless of running speed, a sacral-mounted IMU may be useful for monitoring the rehabilitation stage and return-to-play of an injured person/athlete. By comparing similarity of movement patterns in an injured individual to baseline patterns obtained prior to injury, rehabilitation stage may be evaluated. Furthermore, this technique allows the symmetry of movement patterns and ROMs in all planes to be determined. As mentioned previously, asymmetric patterns could be normal features of an individual runner’s technique or be indicators of current or incipient injuries [9]. The large apparent differences between runners supports the claim by Williams (2007) that identifying individuals’ structural and functional characteristics is important to understand running performance, economy and injury [5]. The ability to characterize runners’ biomechanical patterns can facilitate the development of cluster-specific training programs. Sacral-mounted IMUs are cost-effective, lightweight, and energy-efficient, minimizing disruption to movement. These devices are ideal for field use, potentially capturing every step of a runner’s training, providing a representative measurement that can consider the effects of fatigue, injury and environmental factors on biomechanics. Data can be transmitted and stored online via cloud services, to make real-time remote coaching feasible.

In this study, a range of participants with different running experiences were analyzed and clustered according to their running patterns. Future research should consider larger numbers of runners with a more consistent range of running experience. This may provide pointers to performance, running economy, and injury rates associated with different clusters and biomechanical patterns. Furthermore, longitudinal studies may in future allow the characterization of runners’ biomechanical characteristics throughout their running lifetimes and examine whether these characteristics are stable or change in response to training, aging, injury and footwear selection.

## 5. Conclusions

Inertial measurement units are capable of accurately capturing runners’ sacral displacements and rotations in three dimensions, providing an alternative to camera-based motion capture that is suitable for use in the field. Initial studies indicate that, regardless of running speed, individuals have robust, distinctive movement patterns. This suggests that IMUs may have utility for performance improvement, injury characterization and rehabilitation monitoring in running.

## Figures and Tables

**Figure 1 sensors-25-00315-f001:**
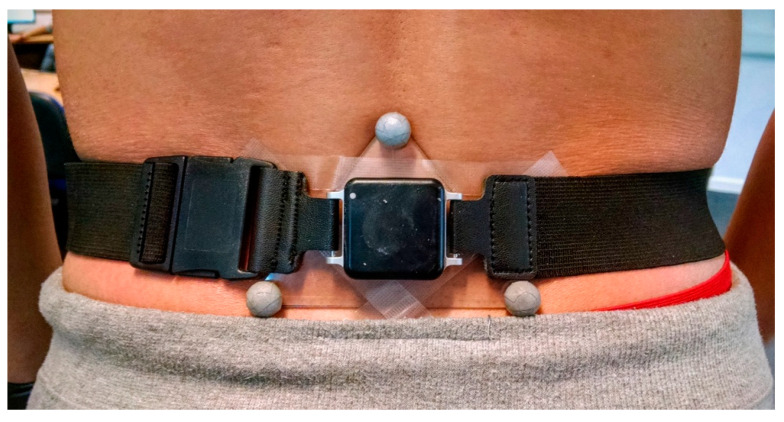
A triangular marker-set attached to a participant’s sacrum using double-sided tape. The IMU was mounted in the middle of the triangle using a belt and flexible bandage.

**Figure 2 sensors-25-00315-f002:**
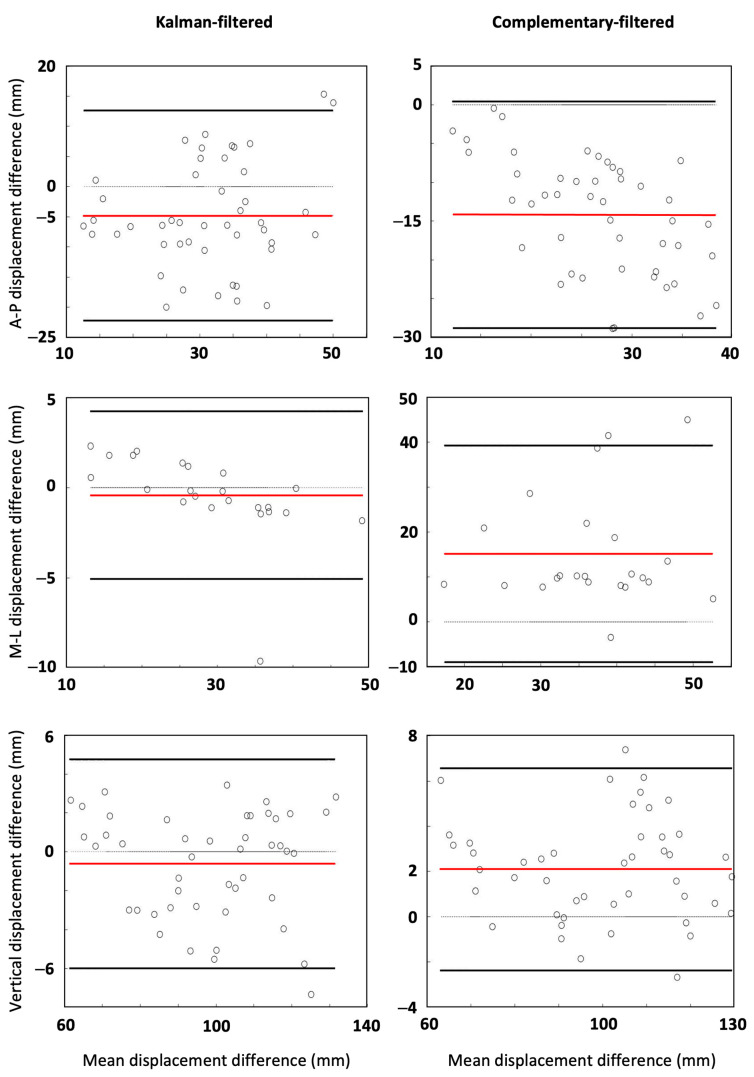
For each filtering method, Bland–Altman plots for ranges of displacements over mean steps of validation trials for anterior–posterior and vertical directions are shown in first and third rows. Bland–Altman plots for ranges of displacements over mean strides of validation trials for medio-lateral direction are shown in second row. In each plot, the red line indicates bias, whereas solid black lines indicate 95% limits of agreement (LOA).

**Figure 3 sensors-25-00315-f003:**
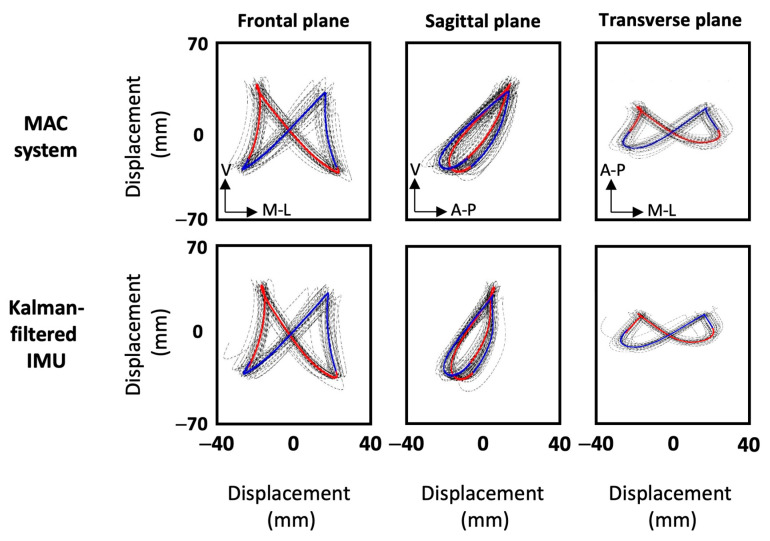
Displacement plots of one participant produced using the motion capture system (first row) and the Kalman-filtered inertial sensor (second row). The plots show frontal, sagittal and transverse planes from left to right. Blue and red traces correspond to left and right steps, respectively.

**Figure 4 sensors-25-00315-f004:**
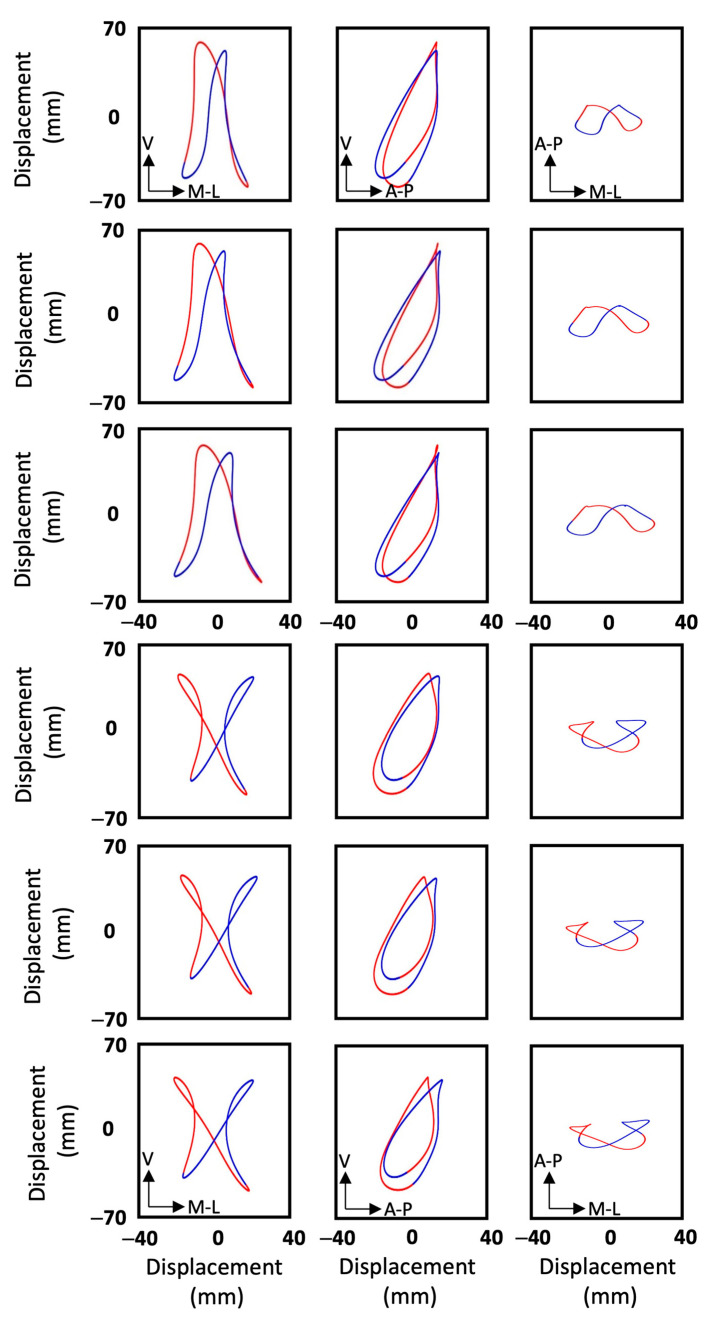
Movement patterns of two participants captured for each of their marathon, 15 km and 5 km running speeds, respectively, from top to bottom. The first three rows are for one participant, while the second three rows are for a second. The plots show frontal, sagittal and transverse planes from left to right. Blue and red traces correspond to left and right steps, respectively.

**Figure 5 sensors-25-00315-f005:**
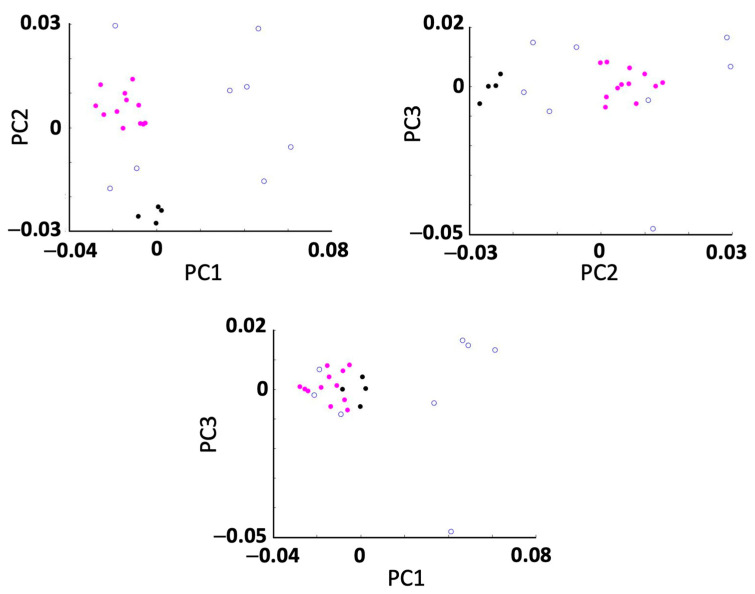
Two-dimensional planes of the first three principal components (PCs) plotted against each other. Magenta and black data points correspond to individuals in clusters one and two, respectively.

**Figure 6 sensors-25-00315-f006:**
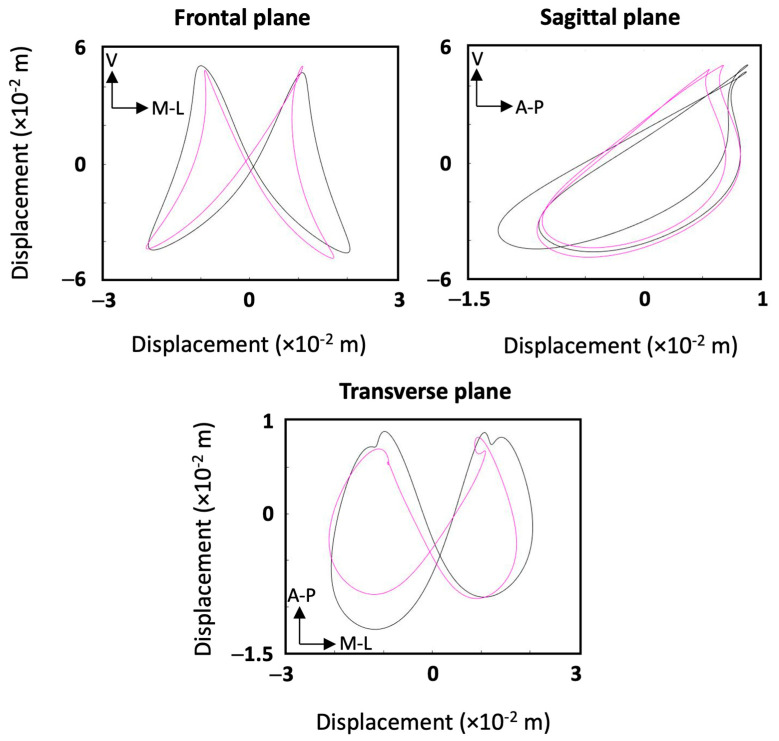
Mean movement patterns of clusters one and two plotted in magenta and black, respectively.

**Figure 7 sensors-25-00315-f007:**
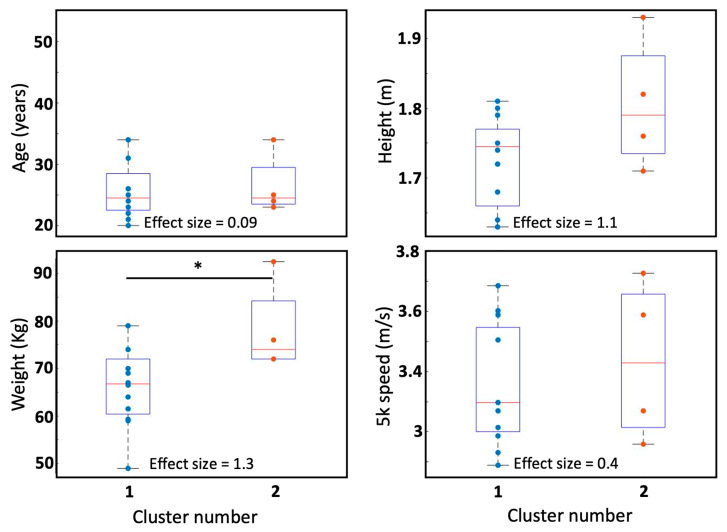
Visual comparison of ages, heights, weights and 5 k speeds between the two clusters using boxplots. Star (*) indicates significant difference (*p* < 0.05). *T*-test was used to compare the two clusters.

**Table 1 sensors-25-00315-t001:** Participant information.

	Participant Numbers	Age	Height (m)	Weight (kg)	Speed Range (km/h)	Mean Running Score	Running Score Range
Validation study							
Male	5	23 ± 1.6	1.74 ± 0.04	70.6 ± 3.7	9.0–18.4	52.5 ± 4.3	46.9 ± 58.6
Female	1	23	1.64	49.0	9.4–13.7		
Movement pattern study							
Male	14	28.2 ± 8.8	1.77 ± 0.07	73.59 ± 8.58	8.8–15.8	54.4 ± 11.3	45.2–72.5
Female	10	24.2 ± 3.1	1.69 ± 0.06	62.73 ± 8.34	7.6–17.1	58.2 ± 9.1	42.3–84.7

**Table 2 sensors-25-00315-t002:** Summary of validation results. Mean difference, limits of agreement and RMSE for measuring acceleration, velocity and displacement using each of the Kalman and complementary filtering methods. Abbreviations: RMSE: root mean square error; LOA: limits of agreement; V: vertical; AP: anterior–posterior; ML: medio-lateral; acc: acceleration; vel: velocity; dis: displacement.

Kalman Filter					
		Mean Difference	LOA	RMSE (Instantaneous)	RMSE (Mean Stride)
	AP acc (m/s^2^)	1.94	−1.72 to 5.61	1.97	1.86
	ML acc (m/s^2^)	0.20	−1.30 to 1.69	0.45	0.38
	V acc (m/s^2^)	0.27	−1.47 to 2.02	0.72	0.67
	AP vel (m/s)	−0.058	−0.330 to 0.215	0.074	0.065
Sacrum	ML vel (m/s)	0.009	−0.038 to 0.056	0.023	0.014
	V vel (m/s)	0.006	−0.009 to 0.110	0.050	0.031
	AP dis (mm)	−4.8	−22.0 to 12.0	5.2	3.7
	ML dis (mm)	−0.4	−5.0 to 4.0	3.1	0.8
	V dis (mm)	−0.6	−6.0 to 4.0	5.8	1.8
	Obliquity (deg)	−0.50	−2.95 to 1.95	0.85	0.77
Pelvis	Tilt (deg)	−0.10	−0.89 to 0.69	1.06	0.69
	Rotation (deg)	−0.49	−2.56 to 1.58	1.54	1.22
**Complementary filter**					
		**Mean Difference**	**LOA**	**RMSE** **(instantaneous)**	**RMSE** **(mean stride)**
	AP acc (m/s^2^)	−0.24	−5.61 to 5.14	2.46	2.36
	ML acc (m/s^2^)	1.91	−3.08 to 6.90	0.81	1.11
	V acc (m/s^2^)	0.41	−0.94 to 1.75	0.95	0.82
	AP vel (m/s)	−0.232	−0.462 to 0	0.124	0.121
Sacrum	ML vel (m/s)	0.121	0.024 to 0.21	0.042	0.072
	V vel (m/s)	0.055	−0.034 to 0.145	0.055	0.036
	AP dis (mm)	−14.2	−28.8 to 0.4	7.9	7.2
	ML dis (mm)	15.2	−8.0 to 39.0	4.3	7.5
	V dis (mm)	2.11	−2.0 to 6.0	5.6	2.2
	Obliquity (deg)	4.68	−11.34 to 20.70	4.46	4.32
Pelvis	Tilt (deg)	−0.58	−1.78 to 0.61	5.25	4.58
	Rotation (deg)	−5.03	−22.91 to 12.86	5.93	5.60

## Data Availability

The raw data supporting the conclusions of this article will be made available by the authors on request.

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
