# Peer review of "Validation and Analysis of Recreational Runners’ Kinematics Obtained from a Sacral IMU"

_sensors, 2025, doi:10.3390/s25020315_

Round 1
Reviewer 1 Report
Comments and Suggestions for Authors
The article tackles a timely and relevant issue in sports biomechanics by exploring the use of sacral-mounted inertial measurement units (IMUs) to study running kinematics. The study demonstrates the accuracy of IMUs as an alternative to the gold-standard motion capture systems. Identifying unique movement patterns among recreational runners and suggesting cluster-specific training programs addresses an important gap in personalised sports analytics and rehabilitation. The research emphasises the field applicability of IMUs, which addresses the limitation of lab-only studies and makes it highly relevant for real-world training scenarios. While previous studies focus on specific kinematic aspects (e.g., knee joint movements or lower limb patterns), this study captures movement across all planes using a sacral-mounted IMU, bridging the gap between complex lab-based methods and portable solutions for field use.
The introduction is comprehensive, offering sufficient background and relevant references, while the research design—comprising validation and movement analysis studies—is appropriate and well-structured. The article provides clear evidence of the Kalman filter's superiority over the complementary filter through Bland-Altman analysis and RMSE comparisons. The conclusions regarding IMUs being viable alternatives to motion capture systems are supported by these findings. The results from PCA and clustering effectively back the claim that individual and cluster-specific patterns can be derived, enabling personalised training or rehabilitation. Figures like displacement plots (Fig. 3) and cluster-specific movement patterns (Fig. 6) support the conclusion that sacral IMUs are suitable for field conditions. The integration of validation and cluster analysis strengthens the argument, providing accuracy and practical insights. The sample size (24 participants, with only 4 in one cluster) limits the generalizability, acknowledged in the discussion. A larger dataset would improve reliability.
The conclusions are well-supported, highlighting the utility of IMUs for performance monitoring and injury prevention. However, there is room for improvement in the optimization of filtering parameters, better contextualization of results within practical applications, and increasing the participant sample size to enhance the generalizability of findings. Additionally, the study could integrate more recent advancements in IMU technology to strengthen its relevance.
The references are mostly appropriate and cover foundational work in biomechanics, running kinematics, and the use of IMUs. Notable strengths include Citing comparative analyses of IMUs and motion capture systems​​. Including studies that emphasize the importance of individualized and cluster-specific approaches in biomechanics​​.
Table 2 provides detailed comparative metrics for the filtering methods, enabling a clear understanding of the Kalman filter's superiority. Additional context on how these values impact practical usability could enhance clarity.
Table 1: While the participant demographics are clear, summarising the diversity in running experience could add value.
Fig. 2: The Bland-Altman plots effectively illustrate agreement levels but are densely packed, potentially overwhelming a reader unfamiliar with such analyses.
Fig. 6: The mean movement patterns per cluster are well-presented and effectively support conclusions about inter-cluster differences.
Fig. 7: Visual comparison through boxplots is helpful; however, explicitly highlighting significance markers (e.g., adding effect sizes) within the figure would improve clarity.
Reviewer 2 Report
Comments and Suggestions for Authors
Dear Authors,
The primary purpose of this study was to validate the use of a sacral-mounted IMU for capturing running kinematics, comparing Kalman and complementary filtering methods. The secondary purpose was to capture sacral kinematics for a wide range of participants, and to investigate if any common patterns (clusters) could be defined.
The authors should present the results more clearly. Despite the statistical analysis performed, the reliability of the motor pattern assessment results is unclear.
The methodology should indicate the value of calculating RMSE and other statistics when using IMU. Table 2 should be provided with an explanation of abbreviations for easy reading.
The list of references should be revised. Many outdated sources are cited. It is doubtful that the results obtained can be compared with publications from ten years ago. The discussion should be based on data from modern studies.
Please note the use of abbreviations.
General comment: The manuscript is not ready for publication as the reliability of the study results needs to be improved.
Round 2
Reviewer 2 Report
Comments and Suggestions for Authors
The manuscript has been significantly improved.